# Applications of Essential Oils and Plant Extracts in Different Industries

**DOI:** 10.3390/molecules27248999

**Published:** 2022-12-16

**Authors:** Parisa Bolouri, Robab Salami, Shaghayegh Kouhi, Masoumeh Kordi, Behnam Asgari Lajayer, Javad Hadian, Tess Astatkie

**Affiliations:** 1Department of Field Crops, Faculty of Agriculture, Ataturk University, 25240 Erzurum, Turkey; 2Department of Genetic and Bioengineering, Yeditepe University, 34755 Istanbul, Turkey; 3Department of Plant Sciences and Biotechnology, Faculty of Life Sciences & Biotechnology, Shahid Beheshti University, Tehran 1983969411, Iran; 4Department of Horticultural Sciences, Faculty of Crop Sciences, Sari Agricultural Sciences and Natural Resources University, Sari 4818168984, Iran; 5Department of Soil Science, Faculty of Agriculture, University of Tabriz, Tabriz 5166616422, Iran; 6Department of Agriculture, University of The Fraser Valley, Abbotsford, BC V2S 7M7, Canada; 7Faculty of Agriculture, Dalhousie University, Truro, NS B2N 5E3, Canada

**Keywords:** essential oils, extraction, natural antimicrobial, plant extracts

## Abstract

Essential oils (EOs) and plant extracts are sources of beneficial chemical compounds that have potential applications in medicine, food, cosmetics, and the agriculture industry. Plant medicines were the only option for preventing and treating mankind’s diseases for centuries. Therefore, plant products are fundamental sources for producing natural drugs. The extraction of the EOs is the first important step in preparing these compounds. Modern extraction methods are effective in the efficient development of these compounds. Moreover, the compounds extracted from plants have natural antimicrobial activity against many spoilage and disease-causing bacteria. Also, the use of plant compounds in cosmetics and hygiene products, in addition to their high marketability, has been helpful for many beauty problems. On the other hand, the agricultural industry has recently shifted more from conventional production systems to authenticated organic production systems, as consumers prefer products without any pesticide and herbicide residues, and certified organic products command higher prices. EOs and plant extracts can be utilized as ingredients in plant antipathogens, biopesticides, and bioherbicides for the agricultural sector. Considering the need and the importance of using EOs and plant extracts in pharmaceutical and other industries, this review paper outlines the different aspects of the applications of these compounds in various sectors.

## 1. Introduction

Essential oils (EOs) are volatile compounds with a slight molecular weight and biological activities synthesized in different plant organs like flowers, buds, leaves, branches, stems, seeds, fruits, woods, roots, etc. [1]. Moreover, EOs contain many active compounds, such as alkaloids, tannins, steroids, glycosides, resins, phenols, volatile oils, and flavonoids [2,3]. Nowadays, EOs are becoming popular natural alternatives to synthetic antioxidants to address consumer concerns about the adverse effects of synthetic antioxidants that have toxic effects on consumers and subsequently lead to various cancers. Therefore, in order to maintain and increase consumers’ health and create food security, new and cheap sources of natural antioxidants are increasingly available [4]. 

EOs and extracts are of great importance in various industries due to the presence of large amounts of volatile, aromatic, and bioactive compounds [5]. Also, these effective compounds have inherent antioxidant and antimicrobial properties and play an essential role in the pharmaceutical, food, agricultural, cosmetic, and health industries [6]. This review provides an updated perspective on the current uses of EOs and plant extracts, and their components, in food, agriculture, and cosmetic products, focusing on their antibacterial, antifungal, herbicidal, etc., properties. 

## 2. The Main Compounds Constituting Essential Oils and Extracts

EOs are volatile substances and dissolve in alcohol and some organic solvents. Terpenes may be linear or cyclic, and depending on the number and how the carbon atoms are connected, different chemicals such as terpenes (with ten carbon atoms), terpenes (with 15 carbon atoms), and diterpenes (with 20 carbon atoms) are created (Figure 1) [3,7,8]. Some of the most important compounds of EOs include eugenol, carvacrol, cinnamic acid, thymol, cinnamaldehyde, citral, and geraniol [9].

The volatile molecules move across the lipids of the bacterial cell membrane due to their hydrophobic nature, disrupt the cell wall structure, and make them more permeable [11]. This change in membrane permeability leads to the leakage of ions and other cellular materials, and ultimately, to cell death [12]. The antibacterial activities of EOs are strongly related to the type of the main compounds [13]. For example, phenolic compounds such as cinnamaldehyde, citral, carvacrol, eugenol, and thymol have significant and strong activities followed by terpenes, while compounds such as ketones (β-myrcene, α-thujone, or geranyl acetate) have weaker activities, and hydrocarbons are almost inactive [14]. The primary active compounds that effectively stop the growth of microorganisms are carvacrol, eugenol, and thymol. They do this by disrupting the cell membrane, which leads to changes in the flow of electrons, protons’ driving force, active transport, and the coagulation of cell contents. Figure 2 displays the different effects of essential oils on microorganisms [15].

The quality of active ingredients in EOs and plant extracts is directly related to the way and the method of extraction. In general, large amounts of expensive organic solvents, such as alcohol, are used in the industrial extraction of plant extracts and some EOs [16]. EOs and plant extracts are composed of several active ingredients, and their extraction time will differ depending on the active ingredient. For example, a hydrodistillation time study conducted by Semerdjieva et al. [17] showed that the yield and composition of EO in *Juniperus virginiana*, *Juniperus sabina*, and *Juniperus excelsa* vary with the distillation time. Accordingly, in *J. virginiana*, the highest content of limonene (43%) was obtained during the first 5 min, and the highest content of safrole (37%) was obtained during the window from 10 to 20 min. In *J. excelsa*, the highest α-pinene (34–36%) was obtained during the first 5 min, limonene (39%) during the first 10 min, and cedrol (50–53%) during the 40 to 240 min range. 

Extraction is the preliminary step for separating the desired natural products from the raw materials. Different extraction methods are used based on the ingredients and the compounds in the EOs and the plant extracts [18]. Distillation, sublimation, and solvent extraction are the most common methods used to extract different oils [19].

The conventional and popular methods for extracting active plant substances are water-based classical methods applied to extract EOs, which include steam distillation and hydrodistillation. These techniques extract EO in a balanced quantity without affecting the quality of the EO. Several techniques with unique special mechanisms are created to ensure better extraction efficiency. Ultrasound extraction, microwave extraction, supercritical liquid extraction, and combined extraction with the help of ultrasound waves are advanced extraction techniques [20]. Herbal extracts are divided into the following two groups: liquid extracts and dry extracts.

### 2.1. Herbal Liquid Extracts

During the liquid extraction process, plant resources such as leaves, roots, or flowers, as well as plant gums, are placed in the vicinity of a specific solvent at a certain temperature, pressure, and heat in the extraction device (extraction reactor), and the majority of the metabolites are taken out. If one wants to use this extract in the food industry, the solvent is separated using an evaporation device. At the end, one will have a total extract containing all of the plant’s active or medicinal substances [21,22]. Some substances in plant extracts are bitter substances, such as flavones, flavonoids, mucilages, saponins, silicic acid, tannins, vitamins, lipids, etc. Flavones and flavonoids exist in free form or in combination with glycosides and chemically belong to phenols. Derivatives of flavones are yellow (‘flavone’ comes from the Latin word ‘flavus’, meaning yellow). Flavones are soluble in the plant cell sap. In general, flavonoids have a softening and moisturizing effect on the skin. Mucilages are carbohydrates with a very complex chemical structure and high molecular weight. These substances are insoluble in alcohol. 

### 2.2. Herbal Dry Extracts

Herbal dry extracts can be obtained in different ways, such as by exposing the liquid cane to the temperature and air of the environment by placing in a spray dryer. When preparing medicinal ointments, tablets, and capsules from herbal or animal sources, dry herbal extracts can be obtained in the same way.

## 3. Application of Active Herbal Ingredients in the Food Industry

Medicinal plants are extensively utilized as flavoring agents in beverages, foods, and dietary supplements. Typically, we have been using these herbs as seasonings and spices in our foods for years because they give a special flavor to foods. Also, with their antioxidant properties, they can play a pivotal role in improving the health of society. Foods’ safety and organoleptic characteristics are prone to microbial spoilage and oxidation change. These reactions reduce the nutritional value of the products due to the loss of their components and impact their safety and organoleptic characteristics [23]. Due to emerging consumer demands and international markets, food distribution and retail systems have changed the storage and transfer time of products. Therefore, it can be said that increasing the shelf life of perishable foods is one of the biggest challenges of the food packaging industry [24]. 

In recent years, antimicrobial and antioxidant compounds have been considered as an alternative for food preservation. Bioactive compounds consist of small amounts of additional nutrients and can modulate one or more metabolic processes that can increase health benefits and enhance nutritional value. Furthermore, due to consumer awareness about the problems of synthetic compounds, human health, and their demand for the use of natural compounds like EOs, natural compounds have been recognized as suitable alternatives to improve the shelf life of perishable foods [25]. 

### 3.1. The Use of Essential Oils to Preserve Food

EOs have been used for their antimicrobial, antifungal [8], and antioxidant [26,27] properties. These compounds can increase food’s shelf life by producing natural preservatives [28]. The beneficial and harmless aspects of these EOs should be considered to ensure the safety of these compounds for using them in foods [29]. Table 1 shows some important medicinal plants that are used as antibacterials in the food industry. 

The antibacterial and antioxidant properties of EOs are due to compounds such as linalool, cinnamaldehyde, thymol, carvacrol, and vanillin, which are acceptable for use in the food industry and increase the shelf life of perishable materials. 

All food products are usually affected by microorganisms except for sterilized food products. However, only a tiny portion of these microorganisms have a favorable effect on improving food taste. Most of them reduce the organoleptic quality of the product and cause food spoilage by producing toxins. Accordingly, the microbial safety of food is one of the major concerns of consumers, regulatory organizations, and food industries worldwide. Therefore, several methods of food preservation are used to control or reduce food spoilage. These methods include freezing, restriction of nutrients, acidification, fermentation, pasteurization, and chemical preservatives [40]. Additionally, new technologies such as modified atmosphere packaging, activated films, non-thermal treatments, drying, enhancement of antimicrobial substances, and radiation slow down the growth of microorganisms [41,42]. Such methods are usually associated with reducing the organoleptic features of foods and decreasing consumer acceptability [43]. 

Most of the antimicrobial substances used are synthetic chemical compounds that slow down the development of microorganisms. These substances include sodium benzoate, potassium sorbate, and nitrites commercially used in fruit juices, dairy products, confectionery, meat, and meat products. Nitrite and nitrate are used in the meat industry to prevent the growth of microorganisms and reduce fat oxidation. Reports have indicated that blue baby syndrome occurs in children due to a large amount of nitrite in their blood [44]. It should be mentioned that many synthetic compounds are also found naturally (benzoic acid in blueberries, sorbic acid in rowanberries, citric acid in lemons, malic acid in apples, tartaric acid in grapes, etc.) [45]. 

Since ancient times, spices have been used for flavoring and as natural preservatives in foods [46]. Due to the high chemical diversity of natural compounds, plant extracts could control the growth of microbes in food products. In addition, they are used in traditional medicine, functional foods, food supplements, and recombinant protein production. The lack of reproducibility of activity is the main barrier to using EOs in foods. Although EOs contain chemical compounds, they have different qualitative and quantitative fluctuations in their content, which greatly affects their biological effectiveness [47]. The pungent aroma of EOs is the major obstacle that limits their use in foods, which can affect their organoleptic properties and taste [43]. Accordingly, the strong flavor of EOs should be minimized by carefully selecting the EO based on the type of food. 

Furthermore, the availability of ingredients and the risk of losing biodiversity are other obstacles to using plant EOs as preservatives [48]. EOs have many bioactive compounds such as terpinol, thujanol, myrtenol, neral, thujone, camphor, and carvone, which indicate significant activities. EOs are generally more active in Gram-positive bacteria due to a peptidoglycan layer on the outer membrane. In Gram-negative bacteria, the outer membrane consists of a double layer of phospholipids which is connected to the inner membrane by lipopolysaccharides. Lipopolysaccharides consist of three basic components—lipid A, a core polysaccharide, and O-side chain—which are responsible for the resistance of Gram-negative bacteria to EOs [14]. In some cases, the combination of two or more compounds produces more activity than when they are used individually, which is called synergistic effects [9]. In general, the value of plants depends on the chemicals present in them, which can affect the microbiological, chemical, and even sensorial quality of foods. These plant chemicals are classified into several categories, including polyphenols, flavonoids, tannins, alkaloids, terpenoids, isothiocyanates, lectins, and polypeptides [49]. The following table (Table 2) indicates a list of plant EOs other than the GRAS group that have antimicrobial activity due to their phytochemical compounds. 

Considering the fact that the use of chemical antibiotic compounds has harmful effects on food health and causes problems such as the emergence of antibiotic-resistant bacterial strains, biologically active plant extracts have a great potential to be used as food preservative agents. 

### 3.2. The Use of Plant Extracts to Preserve Meat and Meat Products

One of the main factors that reduces meat quality is lipid oxidation, which is a complex process and can affect the combination and behavior of meat during food processing. Lipid oxidation leads to the formation of several metabolites that strongly affect meat fiber quality [50]. Generally, these complex reactions cause undesirable changes in the sensory and nutritional characteristics of meat and meat products. Therefore, several antioxidant compounds (both natural and synthetic) have been investigated to reduce or prevent these reactions. For this reason, consumers have recently avoided synthetic antioxidants due to their supposed or possible toxicity [51,52].

**Table 2 molecules-27-08999-t002:** Phytochemical compounds present in some plant EOs.

Plant Name	Scientific Name	The Tissue Used	Phytochemical Compounds	Reference
Viper’s bugloss	*Echium vulgare*	Leaf	Flavonoids, catechol, saponins, steroids	[53]
Onion	*Allium cepa*	Bulb	Saponin, ferulic acid, beta-sitosterol	[54]
Black cumin	*Nigella sativa*	Seed	glycoside, melanin, saponin	[55]
Garlic	*Allium sativum*	Garlic clove	diallyl disulfide, diallyl trisulfide, alkenyl cysteine sulfoxide	[56]
Turmeric	*Curcuma longa*	Rhizome	trans-β-farnesene, α- Zingiberene, β-bisabolene	[57]

Plant extracts have greater antioxidant qualities than artificial antioxidants, which can also help the health status of consumers [58]. Due to their activity, natural antioxidants can generally protect biological cells against oxidation processes caused by reactive oxygen species (ROS) [59]. However, the compounds should not harm the features of the products and also be active in low concentrations if they are to have a chance to be used as natural antioxidants in meat and meat-based products [60]. Additionally, they should not be expensive and should be stable during the process. From the perspective of food technology, the real features of natural antioxidants are deeply influenced by their intrinsic nature, production, and storage conditions (for instance, being exposed to oxygen and light, pH, temperature, and storage time), and the compounds should also be non-toxic in doses higher than normal consumption [61]. In recent years, the meat production industry has mainly focused on the development of healthier meat products [62]. This goal was made possible by using two strategies: reducing undesirable substances and increasing the levels of the desired bioactive substances [63]. 

Polyphenols (such as anthocyanins, tannins, and flavonols) and EOs (mainly composed of terpenes) are bioactive components available in plants that can be used by the meat industry. These combinations can be obtained from different parts of the plant, including its seeds, leaves, and fruits. Black pepper (*Piper nigrum* L.), for instance, is a hot spice that is used all over the world and is considered one of the most valuable spices due to its pleasant effects (pungent taste) [64]. The principal components of the compounds in black pepper EO are 3-Δ-carene, caryophyllene, α-pinene, β-pinene, and limonene [65]. Oregano (*Origanum vulgare* L.) is another flavoring plant in various foods around the world [66]. The most important bioactive compounds in oregano are rosmarinic acid, Phenyl β-D-glucopyranoside, catechin, and luteolin-7-o-rutinoside [67]. Green tea (*Camellia sinensis*) is one of the other valuable plants. It can be consumed as prepared drinks, thick capsules, tablets, and dried leaves for home preparation [64]. Green tea leaves contain bioactive compounds from the family of phenolic compounds, among which epigallocatechin-3-gallate (the main phenolic compound) can be mentioned [68]. Sage (*Salvia officinalis* L.) can be mentioned as another widely used medicinal plant, which is used in food preparation and in medicine [69]. Some of the major compounds of the common sage plant are α-pinene, β-pinene, limonene, 1,8-cineole, and camphene [70]. Pitanga (*Eugenia uniflora* L.) is an indigenous plant of South America that is mainly known for its fruits. However, the leaves of this plant have been primarily considered due to the presence of phenolic compounds; for example, quercetin, myricetin, ellagic acid, isorhamnetin, and quinic acid [71]. The guarana plant is also indigenous to Brazil, and the seeds of this plant are used commercially in the food industry to prepare extracts, syrups, and juices [72]. The phenolic compounds present in guarana seeds include anthocyanins (pelargonidin and 4-ethylcatechol), flavones (luteolin and apigenin), phenolic acids (caffeic, dihydrocaffeic, homovanillic acids), and tyrosols (tyrosol, tyrosol acetate and hydroxytyrosol acetate) [73]. 

The use of natural bioactive compounds to increase shelf life can be mentioned as the uses of plant extracts in meat and related products. This issue is always associated with the effect of these extracts on slowing down oxidative processes, preventing color change, and losing the quality of these products (Table 3). Zhang et al. [74] investigated the effect of EOs extracted from black pepper (0.1 and 0.5%) in the storage of fresh meat (9 days at a temperature of 4 °C). They observed that this EO prevents metmyoglobin formation and reduces the formation of lipid oxidation products after three days of storage. In a study conducted on the use of essences and EOs for the preservation of lamb nuggets (patties), it was found that a combination of EOs and plant oils such as cassia, clove, and thyme resulted in a significant reduction in lipid oxidation during storage (30 days at 4 °C under aerobic condition) [75]. Therefore, using guarana seed extract improves the stability of meat products. Adding guarana seed extract at doses of 250, 500, and 1000 mg/kg leads to the preservation of the color of the hamburger and the reduction of the formation of lipoprotein oxidation products in the beef hamburger [75]. 

Using extracts rich in bioactive compounds from plant leaves can also improve the preservation of meat products. Ramirez-Rojo et al. [78] investigated the effect of mesquite leaf extract on the sensory aspects of pork meat during the storage process. They reported that the progress of meat lipid-protein oxidation was significantly affected by the strong antioxidant activity of this extract. Furthermore, the evaluation of storage stability indicated that the reduction of redness compared to the control sample (without antioxidants) was until the end of the storage period.

Pythanganism leaves are a source of biologically active compounds used to preserve meat products. Studies have indicated that this extract (250, 500, and 1000 mg/kg) in meat storage (18 days at 2 °C) was accompanied by increased protection against lipoprotein oxidation. Additionally, the color change, which is usually observed during the storage of meat products, is a significant indicator of the potent antioxidant potential of this extract [61]. In another investigation conducted with a focus on improving the stability of the appearance and health of meat products during storage, it was discovered that the antioxidant extract from *Moringa oleifera*, which is rich in fiber, prevented lipid oxidation during storage (20 days at a temperature of 4 °C).

Studies indicated that using black cumin ethanol extract to prepare salmon-based marinade reduces the amount of yeast and coliform contamination [79]. Furthermore, Lee et al. [80], reported that adding green tea or rosemary (1 or 3%) to rice cake can significantly reduce the population of *B. cereus* and *S. aureus* in three days of storage at room temperature (22 °C). Turmeric, which contains a combination of curcuminoids plus EOs, is known as a spice and has a vast range of biological activities [81]. Also, another investigation indicated that the use of turmeric extract (1.5% *v*/*v*) alone or in combination with shallot extract (1.5% each, *v*/*v*) preserves the quality of packaged trout stored in a refrigerator for more than 20 days [82].

### 3.3. The Use of Plant Extracts for Food Packaging

Extracts that are rich in bioactive compounds can also be used to improve food packaging features. It should be mentioned that a significant process of active antioxidant packaging for meat replaces artificial additives with natural bioactive compounds [83]. The most typical natural antioxidants employed in these innovative packaging technologies are tocopherols, EOs, and plant extracts such as rosemary, oregano, tea, etc. Accordingly, green tea extract combined in films made from dried protein seeds has been successfully used in the classification of pork, which has reduced lipid oxidation compared to control samples [84]. Studies have shown that a combination of green tea extract in potato starch film significantly improves the mechanical features of meat and reduces the oxidation of packaged fresh beef [85]. In another study, researchers examined the effect of green tea extract on the shelf life of cooked ham (21 days at 2 °C), and reported that the active packaging formulated with green tea extract (1%) was effective in preserving meat color [86]. In another study on lamb ham, it was indicated that the use of oregano extract, without the need for phenol terpene compounds, prevented the oxidation of the bioprotein part compared with the control samples during 120 days of storage at 18 °C [67]. Another study on ham also indicated that the combination of oregano essence and green tea extract (1%) stabilizes the sensory features of cooked ham after 21 days of storage at 2 °C, and results in better surface color change and long-term storage scores [77]. 

## 4. Application of Essential Oils and Plant Extracts in the Agriculture Industry

### 4.1. Application of Essential Oils and Plant Extracts as Insecticides

Plants are a rich source of defensive chemicals. These substances may exhibit insecticidal, repellent, attractant, antinutritional, and growth-regulating effects on insects and usually have minor adverse effects on non-target organisms (parasites and predators) and the environment. Traditionally, synthetic insecticides have been used for many years to control postharvest diseases. Along with the benefits of these insecticides, the expansion of their use has caused consumers to worry about the possible dangers of toxic substances in crop residues. This issue has challenged the usefulness of pesticides and created the need for developing other methods to reduce the usage of chemicals. Controlling and reducing the damage of pathogenic pathogens after harvest using plant products such as plant extracts and essences in recent years could be a suitable alternative to reduce or eliminate the use of chemicals to control postharvest diseases of fruits and vegetables [87]. Currently, there are products based on neem tree extract (neem) and EOs of cloves, rosemary, mint, cinnamon, lemon, thyme, etc., produced commercially for sanitary, agricultural, and greenhouse pest management [88]. 

Plant extracts as biological stimulants and plant protection products play an essential role for modernizing agriculture [89]. Semerdjieva et al. [90] studied the biological activity of four juniper species EOs as biopesticides and their results demonstrated that all tested EOs have significant insecticidal and repellent activity against two aphid species *Sitobion avenae* (English seed aphid) and *Rhopalosiphum padi* (bird oat cherry aphid) at 1%, 2.5% and 5% EO concentrations in solutions. Turmeric (*Curcuma longa*), in addition to the antibacterial properties mentioned above, produces biological activities such as insect repellant and anti-snake venom activity that can be used as bioinsecticide [91]. Also, various studies have shown that *J. virginiana* EO have activities against insects and pathogens [17,92,93]. Yohana et al. [94] studied the important anti-mosquito properties of *Juniperus virginiana* (Cupressaceae) EOs against dominant malaria vectors, and their findings demonstrated that its EO has the potential for the development of new, efficient, safe, and affordable agents for mosquito control. Also, Semerdjieva et al. [95] reported that *Satureja pilosa* EO has larvicidal and mosquito repellent activities against Aedes aegypti that can be utilized for the development of new mosquito management control products. 

### 4.2. Application of Essential Oils and Plant Extracts as Herbicides

A biological method that uses allelochemicals for controlling weeds is an excellent alternative in organic systems. Most allelochemicals are used as secondary metabolites in plants, like EOs, tannins, alkaloids, and glycosides, as alternative strategies for weed management. For example, the findings of Semerdjieva et al. [96] showed that *J. sabina* and *J. exselsa* EOs could manage the control of weed seeds. 

### 4.3. Application of Allelopathy, Essential Oils and Plant Extracts Effects on the Germination of Seeds

The term “allelopathy” in plants is a non-resource-based beneficial or harmful interaction among plants and is more related to allelochemical release [97]. Allelopathy with the help of plant EOs can be practical in modern agriculture. One of its applications in modern agriculture is preventing preharvest sprouting (vivipary) in plants such as wheat and barley [98]. In corn plants, seed germination in the cob head, which can happen due to rain at the time of ripening and before it is harvested due to adverse weather conditions, causes a significant decrease in yield and quality. Of course, preharvest germination depends on many other factors, such as seed maturity stage, dormancy, the cultivar’s genetic resistance, the amount and duration of rainfall, and environmental temperature [99].

Zheljazkov et al. [30] studied the allelopathic effects of EOs from some different plant species, especially hyssop (*Hyssopus officinalis*), lavender (*Lavandula angustifolia*), English thyme (*Thymus vulgaris*), costmary (*Chrysanthemum balsamita*), lovage (*Levisticum officinale*), and cumin (*Cuminum cyminum*), on the seed germination and seedling growth of wheat (*Triticum aestivum*) and barley (*Hordeum vulgare*), and their results can be used in the development of commercial products for controlling the preharvest sprouting of wheat and barley. Their results showed that there are different percentages of EOs in different plants. *Cuminum cyminum* containing 49.6% cumin aldehyde, 10.4% para-cymene, 11.6% α-terpinen-7-al, and 9.1% β-pinene completely inhibited the germination of barley and wheat with the application of 10, 30, and 90 µL, respectively. EOs of *C. balsamita* (43.7% camphor, 32.4% trans-thujone, and 11.6% camphene) and *H. officinalis* (39.8% cis-pinocamphone, 9.8% trans-pinocamphone, 11.4% β-pinene, and 7.5% β-phellandrene (in amounts of 30 and 90 μL) completely inhibited barley and wheat roots per seed, root length, seedling height and germination (%), respectively. The percentage of wheat seed germination was completely inhibited by using *L. angustifolia* and *T. vulgaris* EOs at concentrations of 30 and 90 µL, respectively. Several studies have shown that cumin, hyssop, castor oil, and lavender EOs have the potential for controlling aphids such as *Rhopalosiphum padi* [100]. In a study by Mancini et al. [101], it was observed that EOs of cumin reduce germination percentage from 100 to 1000 mg L^−1^, which led the researchers to conclude that cumin EOs is an allelopathic agent for weed control and should be a promising agent for organic cultivation. *Lavandula angustifolia* EOs was also shown to prevent the emergence of seedlings of the weed species *Amaranthus retroflexus* and *Portulaca oleracea* without much negative effect on tomato [102].

### 4.4. Application of Essential Oils and Plant Extracts as Antibacterial and Antifungal Agents

Bacteria are one of the leading causes of plant diseases and cause damage to many agricultural products. These plant pathogens can reduce vegetables’ shelf life and cause significant economic losses for farmers and producers of vegetable products. In recent years, the indiscriminate use of chemical bactericides and antibiotics to control plant diseases has caused resistance in pathogenic bacteria. For this reason, finding compounds of natural origin has been considered an effective and safe approach to restrain the growth of bacteria and improve the shelf life of foods. Due to the presence of antimicrobial compounds, plant extracts have a high potential to be used as natural preservatives and can be used to control decay and raise the shelf life of plant products. Therefore, considering the effectiveness of medicinal plants in eliminating disease agents and diseases, these plants can also play a remarkable role in eliminating monocytes and insects that cause agricultural damage. 

Semerdjieva et al. [103], when studying juniper tree species (*J. oxycedrus* L.), showed that EOs extracted by water distillation through standard Clevenger apparatus (Clevenger) and semi-commercial steam distillation (SCom) have balanced activity against selected pathogens (*Botrytis cinerea*, *Fusarium* spp., *Rhizoctonia solani*, *Colletotrichum* spp., and *Cylindrocarpon pauciseptatum*). Table 4 shows a list of plants and combinations of effective substances with antipathogen properties.

## 5. Application of Essential Oils and Plant Extracts in the Cosmetics Industry 

The use of EOs and herbal extracts as cosmetics has attracted peoples’ attention for a long time, and active substances obtained from plants were used as perfumes or cosmetics. Historical findings obtained from the countries of origin of these plants, such as India, China, Egypt, and Iran, confirm the long-time usage of medicinal plants as cosmetic products. 

It is estimated that more than 80% of the world’s population uses conventional medicine (plant extracts or their active compounds) to meet their health needs [111]. For example, in ancient Egypt, the oil of medicinal plants was extracted by steaming, while the Romans and Greeks used the distillation method for extraction. Also, with the advent of Islamic civilization, the use of medicinal plants accelerated with innovative approaches, and the methods of extracting oil, EO, and their substance took a practical step in the path of evolution. In addition, the EO obtained from species like *Cedrus libani*, *Ocimum kilimandscharicum*, *Artemisia annua*, *Acacia vestita*, *Piper angustifolium*, *Sassafras albidum*, and *Rosmarinus officinalis* contains camphor, which has been identified as an aromatic substance for centuries, and in ancient Japan and China. In other Asian and European countries, it has been used medicinally as well as in culinary and cosmetic applications [112,113]. 

Currently, consumers’ tendency to purchase herbal cosmetics, which is both environmentally friendly and renewable, has increased [114]. In recent years, “natural cosmetics” recorded a large quota of cosmetics (about $40 billion by 2021, which is about 10% of the global cosmetics market) [115]. Due to their anti-inflammatory, antimicrobial, and antioxidant properties, plant EOs, either as active ingredients or as preservatives, are used in various cosmetic products such as moisturizers, lotions, and cleansers in skin care. Cosmetics, conditioners, masks, or anti-dandruff products are used in hair care products, lipsticks or perfumes in perfumery [115].

By adding rosemary and chamomile EOs to shampoo, EOs can quickly penetrate the scalp, feed the hair follicles, moisturize the hair, strengthen the hair, and eliminate the adverse molecules that block the pores of the skin [116]. EOs have a major impact on stimulating hair growth and preventing hair loss [117]. It was reported that topical mint oil extracted from Mentha piperita with a relatively low dose (3% weight on weight) is typically used to stimulate hair growth [118]. One of the most common applications of EOs in skin care is to prevent acne from appearing, using the ability of EOs to inhibit *Propionibacterium acnes*, which Citronella’s EO has the ability to do [119]. EOs can be used in sunscreen creams because they can absorb most ultraviolet rays (in the wavelength range of 290 to 400 nm), prevent aging, sunburn, wrinkles, and other skin damage [120]. A cream formulation that contains EOs of *Calendula officinalis* was studied in laboratory conditions, and the results confirmed that the prepared formula had good properties for protecting the skin from exposure to sunlight [121]. 

Preservatives are added to cosmetic products to prevent microbial spoilage and thus increase the shelf life of the products. It is also necessary to protect the consumer against potential infections. Although chemical preservatives prevent microbial growth, their safety is questioned by consumers. Therefore, there is a considerable interest in producing preservative-free or self-preservative cosmetics [122]. Therefore, the use of essences and extracts derived from plants in the production of cosmetics as multipurpose antimicrobial compounds, both as an alternative or natural preservatives and with anti-pathogenic properties, can help to improve the quality of these products. Some species from Nepta, for example *Nepeta cataria var* (Citriodora), *Nepeta cataria*, and *Nepeta grandiflora*, are used in herbal cosmetics. In these three species, the major compounds were nepetalactone and geraniol in *N. cataria*; citronellol and geraniol in *N. Citriodora*; and o-cymene, c-terpinene, carvacrol and p-cymene in *N. Grandiflora* [123]. Table 5 shows a list of plants and combinations of effective substances with antipathogen properties for use in the cosmetic and health industries. 

## 6. Conclusions and Future Perspectives

EOs and extracts contain important secondary metabolites produced from many plant species and can be used as rich sources for the preparation of herbal medicines or as preservatives and antioxidants in pharmaceutical, food, agricultural, and cosmetic products. Also, due to the presence of various active ingredients, they have antibacterial, antifungal, antiviral, insecticidal, and allelopathic properties, and these properties can be used to improve the quality of the products, to increase the shelf life of the products, and to protect plants from pests and plant diseases. However, since the safety of chemical preservatives in the food, cosmetics, and agricultural industries has been questioned, more studies are necessary to evaluate EOs and plant extracts as alternatives for use in various industries. Furthermore, due to the presence of different bioactive compounds in EOs and extracts, it is necessary to check the compounds in them. Efforts need to be directed towards the use of automation and high throughput screening to search for novel bioactivities of EOs. Also, the huge amount of information generated by in vitro assays must be confirmed by vivo assays and large-scale clinical investigations.

## Figures and Tables

**Figure 1 molecules-27-08999-f001:**
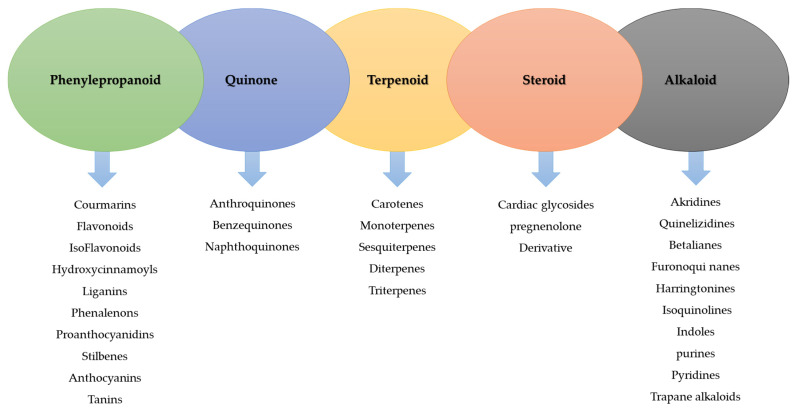
Classification of the active ingredients of medicinal plants (modified from Sidhu [10]).

**Figure 2 molecules-27-08999-f002:**
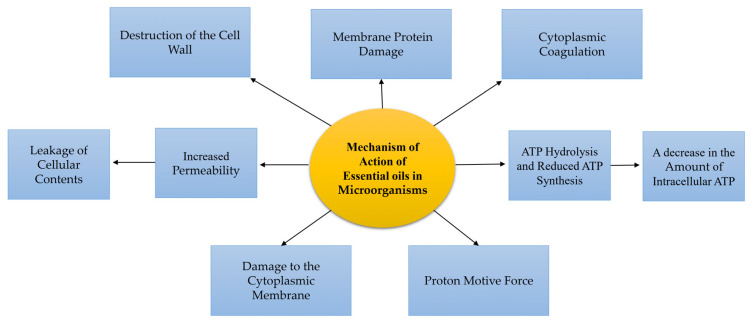
Various types of the mechanisms of the activities of EOs on microorganisms.

**Table 1 molecules-27-08999-t001:** EOs and plant extracts with antimicrobial properties that are used in the food industry.

The Scientific Name	Effective Compounds in%	Microorganisms	Reference
*Lavandula angustifolia*	47% linalool acetate, 28.4% linalool	*Aeromonas caviae* *Pseudomonas aeruginosa* *Salmonella enterica*	[30]
*Origanum* *vulgare*	38.2% para-cymene, 25.6% thymol, and 13.6% γ-terpinene)	*Clostridium botulinum spores*	[30]
*Salvia* *rosmarinus*	limonene (6.23), camphene (6.0), and linalool (5.7)	*Listeria monocytogenes*	[31]
*Thymus* *serpyllum*	carvacrol (0.2–0.6)	*Salmonella enteritidis* *Salmonella typhimurium*, *Escherichia coli* serovar *Yersinia enterocolitica* serotype	[32]
*Cymbopogon* *citratus*	geranial (33.3), limonene (5.8), and geranyl acetate	*Staphylococcus aureus*, *Escherichia coli*, *Candida albicans*, *Bacillus cereus* and *Salmonella* *typhimurium*	[33]
*Eucalyptus* *camaldulensis*	cineole (eucalyptol) (80–90)	*Listeria monocytogenes*, *Staphylococcus aureus*, *Bacillus cereus*, *Salmonella typhimurium*, and *Esherichia coli*	[34]
*Mentha* *pulegium*	Mentha, 1,8-cineole	*Staphylococcus aureus* and *Escherichia coli*	[35]
*Salvia* *officinalis*	α, β-tuyon, 1,8-sineol, kamfor	*Staphylococcus aureus* and *Streptococcus*, *Candida* *albicans*	[36]
*Helichrysum* *italicum*	1,8-cineole, α-copaene, (E)-β-ionone, γ- cadinene, selina-3,7(11)-diene, epi-α-cadinol, α-cadinol, octadecane, isophytol and tricosane	*Staphylococcus aureus*, *Bacillus subtilis*, *Aspergillus brasiliensis*,	[37]
*Curcuma longa*	α-Turmerone (35.16), ar-Turmerone (25.47), and Curlone (18.19)	*S. aureus*	[38]
*Juniperus* *oxycedrus*	α-pinene, limonene, α-curcumene, γ-cadinene, δ-cadinene, manoyl oxidearyophyllene, α-*caryophyllene*, *caryophyllene oxide*	*Yersinia enterocolitica*, *Staphylococcus aureus subsp. aureus*, *Enterococcus faecalis*, *Streptococcusp neumoniae*, *Candida kruseii*, and *C. tropicalis*.	[39]

**Table 3 molecules-27-08999-t003:** List of uses of plant extracts as natural antioxidants in meat and meat products.

Herbal Essential Oil	Target Product	Amount Consumed	Time and Conditions of Storage	Result	Reference
Black pepper	Fresh meat	0.1–0.5%	9 days at 4 °C	B and C	[61]
Guarana seed	Lamb burger	250 mg/kg	18 days at 2 °C	B and C	[76]
Mesquite leaf	Minced meat	0.1–0.05%	10 days at 4 °C	A and C	[73]
Moringa	Chicken nuggets	1–2%	20 days at 4 °C	A, C and D	[75]
Green tea	Chopped cooked ham	1%	21 days at 2 °C	C	[77]
Oregano	Lamb burger	13.32, 17.79, 24.01 mL/kg	120 days at 18 °C	A and C	[67]
	Chopped cooked ham	1%		D	[77]

A = lipid oxidation reduction; B = lipoprotein oxidation reduction; C = reduction of color oxidation; D = decrease sensory degradation.

**Table 4 molecules-27-08999-t004:** Some medicinal plants with anti-pathogenic properties that are used in the agricultural industry.

Scientific Name of the Plant	Compounds	Pathogen Name	Reference
*Artemisia persica*	Artemisinin	*Cochliobolus* *sativus*	[104]
*Zingiber officinale*	Gingerburn and oleoresin	*Sclerotinia sclerotiorum*	[105]
*Tagetes patula*	Quercetagetin; quercetin; patuletin; quercetin-3-glucoside	*Meloidogyne incognita*	[106]
*Thymus vulgaris*	Thymol and carvacrol	*Bemisia tabaci*	[107]
		*Erwinia Amylovora*	
*Lavandula angustifolia*	linalol, linalil	*Alternaria alternate*	[108]
*Pimpinella anisum*	Anetol	*Aspergillus flavus*, *Phoma sorghina*, *Alternaria alternata*, *Botrytis cinerea*	[109]
*Nigella sativa*	Carvon		
*Juniper galbulid*	β-Elemene, γ-Elemene, and τ-Muurolol	*lostridium perfringens*, *Juniperus**communis* against *Candida clabrata*, and *Juniperus oxycedrus* against *Staphylococcus aureus*	[110]

**Table 5 molecules-27-08999-t005:** Some medicinal plants used in the cosmetic industry.

Scientific Name of the Plant	Compounds in%	Function	Reference
*Matricaria chamomilla*	Camazolin, alpha bisabolol	As natural cosmetic preservatives in cream	[124]
*Calendula officinalis*	Flavonol glycoside and beta-carotene (30–40)		
*Lavandulla officinallis*	linalol (35–55), linalyl		
*Aloe vera*	Antrakinonlar	As natural cosmetic preservatives in cream	[125]
*Cannabis sativa*	Cannabidiol	Skin anti infections	[126,127]
*Urtica dioica*	Polyphenol	As strong cleaning in shampoos	[128]
*Glycyrrhiza glabra*	Glycyrrhizin	Anti-inflammatory of skin in cream	[129]
*Juniper galbuli*	β-Elemene, γ- Elemene, τ-Muurolol	Anti-inflammatory of skin in cream	[110]

## Data Availability

Not applicable.

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
