# Peer review of "Applications of Essential Oils and Plant Extracts in Different Industries"

_molecules, 2022, doi:10.3390/molecules27248999_

Round 1

Reviewer 1 Report

When I read this article, I got the impression that it is disorganised, negligent, and prepared in a rush.

There are parts of the text that are incorrect and can mislead the reader, as an example, the very first sentence from the Introduction: ‘Essential oils and plant extracts are secondary metabolites from plant sources and have biological activities’ or line 233: ‘Natural compounds, such as plant extracts (...)’ – plant extracts are not secondary metabolites, they contain active compounds. Another example, line 187: ‘Active substances are compounds that increase the shelf life of food ingredients that results in improving and preserving food quality. These compounds are divided into two categories: natural and kinetic.’ – kinetic compounds? I assume that the authors meant synthetic. Furthermore, this is not the definition of active compounds. One more example, line 254: ‘In some cases, the combination of one or two compounds indicates more activity than the time they are used individually. Accordingly, synergistic effects are observed when they are used in combination [32].’ – I think it should be: two or more? The second sentence is just a repetition of the first one. Such errors are unacceptable and these are only a few examples of the whole manuscript.

There are many repetitions of words and sentences, for example, line 192: ‘Essential oils (EOs) are the secondary plant metabolites (…). Also, EOs have been used for for many years for their antimicrobial and anti-fungal properties [31] and antioxidant [50,51] properties.’ – apart from the repetitions in this one sentence, this information also appears in further parts of the manuscript, for example:

Line 35: ‘Essential oils and plant extracts are secondary metabolites from plant sources and have biological activities’

Line 43: ‘Medicinal plants are important in various industries due to their large amounts of volatile aromatic and bioactive compounds’ and two lines later, line 45: ‘These volatile compounds have inherent antioxidant and antimicrobial properties (...)’

Line 98: ‘Essential oils are volatile compounds’ and two lines later, Line 100: ‘Essential oils are volatile substances’

Line 109: ‘As previously mentioned, essential oils obtained from medicinal plants play an important role as antimicrobial agents.’

Line 198: ‘The biological properties of EOs include antibacterial, antifungal, and  antioxidant properties.’

Line 204: ‘EOs and their compounds (…) are acceptable for use in the food industry and increase the shelf life of perishable materials due to their properties such as antimicrobial, antifungal, and antioxidant activity.’

Line 231: ‘The antimicrobial and antioxidant features of plants are related to EOs and their secondary metabolites (…)’

Line 252: ‘Furthermore, the major compounds available in EOs and the features of volatile compounds and their interactions may increase the antibacterial activity.’

Line 458: ‘Furthermore, EOs often prevent the growth of bacteria and fungi.’

Line 195: ‘Currently, these compounds are receiving more attention because they are considered as harmless to human health [49]’ and 4 lines later, line 199: “Therefore, the beneficial and harmless aspects of these EOs should be considered to ensure the safety of these compounds for using them in foods [52].’ Line 228: ‘Studies indicate that the usage of natural antimicrobial substances such as organic acids, essential oils, plant extracts, and bacteriocins can be a suitable alternative to provide food safety [57].’ Line 265: ‘(...) the usage of medicinal plant extracts can be a suitable, cost-effective and safe to control microbial activities in the food industry’.

Line 441: ‘Medicinal plants, as cosmetics, have been of particular interest to people for a long time.’ Line 443: ‘Oils and active substances obtained from plants are used as perfumes or cosmetics.’ Line 445: ‘(...) the basis of the use of medicinal plants as cosmetic products goes back thousands of years.’

The same trend applies to descriptions of active compounds and plant extracts. The whole manuscript consists of a cluster of separate sentences. The language should be improved and the article should be rewritten (one issue should be discussed in one section) to form a coherent whole. The information about the solvents and extraction methods should be included in a separate section. The additional solvent types and extraction techniques should be mentioned – their pros and cons, extraction efficiency, etc. Different forms of the use of essential oils/active compounds should be added.

The part about the use of active compounds in pharmacy or medicine was treated very negligently – this aspect was discussed in three lines based on the example of Covid 19; line 479: ‘Although the importance of EOs and herbal extracts in the treatment and prevention of diseases is not hidden from anyone, considering the 2019 Covid epidemic, a brief review of the research conducted on the effect of EOs and herbal extracts is shown in Table 6.’

Lines 60-88 – should not be a part of the Introduction section.

The whole conclusion section should be rewritten.

Not all Latin plant names are in italics and they are written in an unordered manner.

The diagrams are prepared in a careless manner in a very poor quality.

A review article should carefully review most of the available sources. The authors listed only a few examples in each section.

The article has potential, but still needs a lot of work and is not suitable for publication in Molecules journal.

Reviewer 2 Report

Congratulations authors, the manuscript has merits and addresses a relevant topic. The work presents an interesting discussion and will add more to current knowledge. The data, as well as the analyzes carried out during the review, are presented in an appropriate way. I suggest that the work be accepted for publication in the journal, but first, authors should carefully read the entire text for corrections of terms and phrases, as well as concordances. I emphasize that the work is well presented, requiring only a few adjustments for its improvement. Also check that all citations are indicated in the references and vice versa.

Reviewer 3 Report

This review paper outlines the different aspects of the essential oils and extracts isolated from medicinal and spicy plants as well as their applications in different industries. The literature used was published in the past decade. There are some technical mistakes (indicated in the .pdf form) that should be corrected before the acceptance the manuscript. 

Reviewer 4 Report

The article titled: Applications of Essential Oils and Plant Extracts in Different Industries presents valuable information. However, the article's goal is too broad, and it is not shown enough. Using essential oils/plant extracts in industries is very common - perhaps separate reviews could be prepared to describe it. Thus, to be complete, the article demands a lot of essential amendments, and the Authors should improve the majority of the article parts. The Authors should re-edit the manuscript. Below I present my comments.

1. Introduction: The phrase: "Essential oils and plant extracts are secondary metabolites from plant sources and have biological activities" is not substantively correct. Neither Essential oils nor plant extracts are not secondary metabolites. They contain secondary metabolites. The authors must correct it. The second sentence: "These advantages make them be used in various industries such as food, agriculture and cosmetic industries in addition to pharmaceutical industries [4,5]." It is not well comprehensible for me (which advantages ?). The reference is needed: "These volatile compounds have inherent antioxidants against diseases caused by microorganisms." I do not understand why the Authors focused on antimicrobial activity – it should not be the primary goal of this article.

There is also a lack the aim in the Introduction.

In my opinion, the Authors should rewrite the Introduction.

2. Paragraph 2 is strange. The Authors give a general title and describe the different aspects of plant compounds – extraction and biological activity (why only antimicrobial is defined very precisely?). In my opinion, too much information is presented in the same part. In Figure 3, why is the scheme of pressurized liquid extraction introduced – it this the only method existing?

3. The paragraph 3 should also contain parts like 3.1. The use of essential oils in the food industry (or similar). Too much information is resented in the introductory part of paragraph 3. However, the information about EO from part "3.1. The use of plant extracts to preserve meat and meat products" should be considered when preparing the part about the EO.

4. I suggest using "essential oils" instead of EO in the titles and subtitles of paragraphs.

5. The information included in paragraph 4 (for example, 4.3.) is too general 4.3. The authors mention the interesting studies by Zheljazkov, but the results of these studies are not known.

6. Paragraph 5 – some Latin names are not italics; more scientific data are also needed because the paragraph is too overall in this form.

7. The COVID-19 part is not necessary – it is incomplete. If the authors decide to publish this, the section must be developed.

8. Authors should improve the quality of the Figures. For example figure 1 – the circles are not the same size, "anthosyanin" – is wrongly written; Figure 2 is also not very symmetrical; 

9. Table 1 – requires refining the details (italic or not; L. etc. or not with the species); not "Scientific name of plants" but "Latin name of plants"; The Lavandula angustifolia and Origanum vulgare – the names of the compound of the essential oils is not well suited, and it is not well known which compound is the ingredient of each essential oils. The same concerns table 2. The column "Results" in the Table 3 is not understandable. Table 4 must also be revised (sometimes the Latin name is mentioned as the first but in the other line, the familiar name); Table 5 – what does it mean "Pathogen name", when it comes to the content of Table 5 – why authors focused on the microorganisms in the subject of cosmetical using?

10. Abbreviations: If Authors once introduced the abbreviation (EOs), they should be continued using this in the text. However, in the text, the Authors use EOs and essential oils. In addition, in the Table captures, the Authors should use the whole "essential oils" name.

11. Conclusion: The first phrase, in conclusion, is: "Plant extracts with antibacterial activity are useful to improve the quality of the products, to increase the products' shelf life, and simultaneously to prevent economic losses." However, the title of this manuscript is: "Applications of Essential Oils and Plant Extracts in Different Industries." It suggests that the manuscript's main aim is to review the antibacterial activity. This is not consistent with the title.

Round 2

Reviewer 1 Report

The authors took the time to improve the article.

Reviewer 4 Report

Thank you to the authors for the amendments introduced to the manuscript. I find that in this form, the article can be published.